# Multi-target Parallel Drug Discovery with Multi-agent Orchestration

**Multiple AI Agent Systems**
Powered by Gemini 2.5 Pro
and Gemini 2.5 Flash-Lite

**Fuad Al Abir**
Department of Biomedical Informatics and Data Science
University of Alabama at Birmingham
Birmingham, AL 35294
fuad021@uab.edu

**Sixue Zhang**
Systems Pharmacology AI Research Center
University of Alabama at Birmingham
Birmingham, AL 35294
szhang66@uab.edu

**Jake Y. Chen**
Systems Pharmacology AI Research Center*
University of Alabama at Birmingham
Birmingham, AL 35294
jakechen@uab.edu

## Abstract

The immense cost and high failure rates of traditional drug discovery necessitate a shift towards more systematic, automated approaches. While specialized AI tools have excelled at individual tasks, their integration into a cohesive, end-to-end discovery workflow remains a significant challenge. We introduce a modular, multi-agent framework that autonomously navigates the early-stage drug discovery pipeline, from target identification to the generation of optimized hit candidates. Our system orchestrates specialized agents that synergize AI Agents for literature mining and generative chemistry with robust machine learning classifiers for bioactivity and ADME/Tox prediction. We demonstrate the system's capabilities by applying it to Alzheimer's Disease, identifying and generating novel inhibitors for five protein targets—SGLT2, CGAS, SEH, HDAC, and DYRK1A, and successfully generated novel molecular scaffolds with high predicted potency and favorable drug-like properties for four of the targets. The framework's failure to build a reliable model for the data-scarce target, CGAS, highlights a key limitation: the performance of autonomous systems is fundamentally tethered to the quality and availability of the underlying data. Our work presents a transparent blueprint for an integrated discovery engine and provides a realistic perspective on the current capabilities of AI agents in science, suggesting they operate most effectively within a human-in-the-loop paradigm where expert oversight guides data curation and model validation. The code is publicly available at https://github.com/UAB-SPARC/agentic-drug-discovery.

## 1 Introduction

The journey of developing a new therapeutic, from initial hypothesis to market approval, is an undertaking of immense cost, time, and risk. Estimates suggest that bringing a single new drug to patients can take over a decade and cost upwards of $2.5 billion, with failure rates exceeding 90%, particularly in the early stages of discovery [7]. These staggering figures underscore a critical bottleneck in modern medicine: our capacity to discover and validate novel therapeutic interventions struggles to keep pace with our growing understanding of disease biology. This challenge is particularly acute in complex neurodegenerative disorders like Alzheimer's Disease, where novel therapeutic strategies

---

*https://www.smartdrugdiscovery.org/

are desperately needed. Moreover, The complexity of navigating vast chemical and biological spaces necessitates a paradigm shift from traditional, often serendipitous discovery methods towards more systematic, data-driven, and automated approaches.

Artificial intelligence (AI) and machine learning (ML) have emerged as powerful catalysts poised to revolutionize this paradigm [8]. AI-driven techniques are now being applied across the entire discovery pipeline, from identifying novel protein targets using natural language processing on biomedical literature [14] to designing novel molecular entities with bespoke properties using deep generative models [9]. Similarly, ML models, particularly gradient boosting algorithms like XGBoost, have become instrumental in predicting critical drug-like properties, such as absorption, distribution, metabolism, excretion, and toxicity (ADME/Tox), enabling the early deselection of candidates destined for failure [4].

However, a significant gap persists between the development of these powerful, specialized AI tools and their integration into a cohesive, end-to-end workflow that can autonomously navigate the early discovery process. While several industrial platforms have demonstrated success, they often operate as proprietary *black-boxes*, limiting broader academic and scientific insight. The challenge lies in orchestrating these disparate components—literature mining, target validation, molecular generation, and multi-property evaluation—into a synergistic system that can reason, plan, and execute complex scientific tasks.

This paper introduces a modular, multi-agent framework designed to address this integration challenge. We present a system composed of specialized AI agents that collaborate to execute an autonomous drug discovery pipeline, spanning from target identification in scientific abstracts to the *de novo* generation of optimized lead candidates. Our framework synergizes large language models (LLMs) for tasks like target mining and molecular generation with robust ML classifiers for bioactivity and ADME/Tox prediction. We demonstrate the system's capabilities by applying it to five distinct protein targets—SGLT2, CGAS, SEH, HDAC, and DYRK1A—and critically evaluate both its successes and its limitations. Furthermore, we explore a parallel agentic framework for the automated generation of the scientific manuscript itself, showcasing how these technologies can accelerate not only discovery but also the dissemination of scientific knowledge. Our work provides a transparent blueprint for an integrated discovery engine and offers a realistic perspective on the current capabilities and near-term challenges—particularly data scarcity and model reliability—that define the frontier of AI-driven science.

## 2   Related Works

Our research is situated at the confluence of three rapidly advancing domains: AI-driven drug discovery, *de novo* molecular design, and multi-agent systems for scientific automation.

**AI-Powered Drug Discovery Platforms**   The application of AI to streamline drug discovery has given rise to several successful integrated platforms. Companies like Insilico Medicine have pioneered an end-to-end approach, utilizing their Pharma.AI platform, which combines the target discovery engine PandaOmics with the generative chemistry engine Chemistry42 to move from hypothesis to a preclinical candidate in under 30 months [18]. BenevolentAI leverages a vast, curated knowledge graph constructed from biomedical literature and databases, which its AI tools query to uncover novel relationships between genes, diseases, and drugs, thereby generating and prioritizing new therapeutic hypotheses [19]. In contrast, platforms from companies like *Schrödinger* place a strong emphasis on physics-based modeling, using technologies such as Free Energy Perturbation (FEP+) to achieve high-accuracy predictions of protein-ligand binding affinities, which is especially critical during lead optimization [20]. Another distinct approach is taken by Recursion Pharmaceuticals, which employs "phenomics"—high-throughput, image-based screening of cellular perturbations—to build massive "Maps of Biology" that guide discovery irrespective of specific molecular targets [21]. While these platforms demonstrate the power of integrated AI, our work differs by proposing an explicit and transparent multi-agent architecture, offering a modular framework that can be understood and adapted by the broader research community.

**Generative AI for *De Novo* Molecular Design**   The core of modern hit generation is *de novo* molecular design, a field recently transformed by deep learning. Early computational methods relied on fragment-based growth or evolutionary algorithms. The advent of deep generative models,

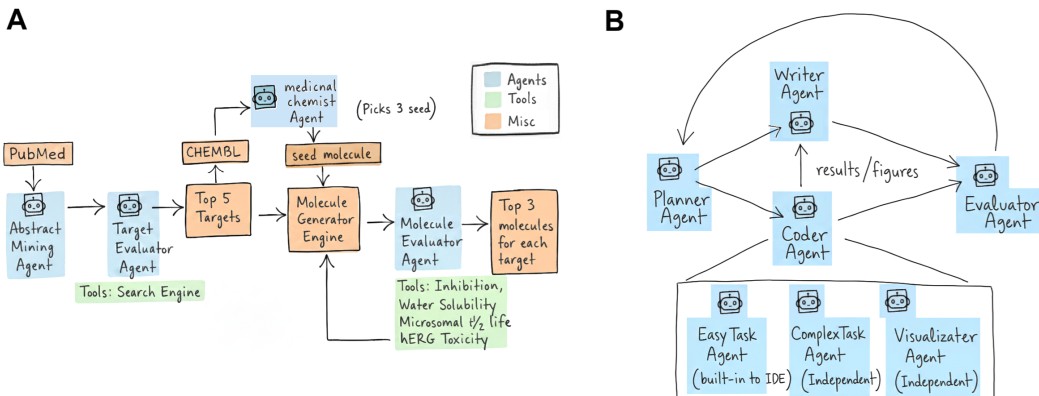

Figure 1: **Agentic Framework for Multi-target Drug Discovery and Scientific Writing. A. The autonomous drug discovery workflow.** The process begins with an Abstract Mining Agent identifying targets from PubMed, which are then scored by a Target Evaluator Agent. Seed molecules for validated targets are selected from ChEMBL. The Molecule Generator (NVIDIA MolMIM) creates novel compounds, which are assessed by the Molecule Evaluator via a sequential filtering cascade (hERG, solubility, etc.). This iterative loop results in optimized inhibitors for each target. **B. The manuscript generation framework**. A Planner Agent orchestrates the process, assigning tasks to a Writer Agent (prose), a hierarchical Coder Agent (code and figures), and an Evaluator Agent (quality control). This conceptual workflow was simulated through manual, iterative prompts to the specified models to generate this paper. The diagram itself was produced by providing a hand-drawn sketch as a prompt to the *nano-banana* image generation model.

such as Variational Autoencoders (VAEs), Generative Adversarial Networks (GANs), and Flow-based models, enabled the design of novel molecules by learning a continuous latent representation of chemical space [13]. More recently, the success of large-scale Transformer models in natural language processing has been adapted to chemistry. Models like NVIDIA's MolMIM [5], used in our framework, treat molecular generation as a language modeling task, allowing them to generate diverse and valid chemical structures conditioned on a starting seed molecule or desired properties. This approach excels at "scaffold hopping"—producing structurally novel molecules that retain key pharmacophoric features, a crucial capability for escaping patent-protected chemical space and discovering new intellectual property.

**Multi-Agent Systems for Scientific Discovery**   While the individual AI components are maturing, their orchestration into autonomous systems remains a key challenge. Multi-Agent Systems (MAS) have emerged as a compelling paradigm for this task [16]. In a MAS, independent, specialized agents communicate and collaborate to solve problems that are beyond the capabilities of any single agent. This architecture mirrors the human scientific process, where experts with different skills (e.g., biologist, chemist, data scientist) work together. Recent conceptual work has proposed agentic AI for various scientific domains, including materials discovery and autonomous experimentation [17]. Our framework contributes a concrete implementation of this paradigm for the early drug discovery workflow. Unlike monolithic platforms, our agent-based approach provides modularity and transparency, where each agent's function—from the **Abstract Mining Agent** querying PubMed [1] to the **Molecule Evaluation Agent** applying a filtering cascade—is distinct and interpretable. By orchestrating these agents, our system automates the complex cycle of hypothesis, design, and evaluation, representing a practical step towards truly autonomous scientific discovery.

## 3   Methodology

Our research introduces a multi-agent framework designed to accelerate early-stage drug discovery, from target identification to lead generation. The framework is composed of two primary workflows executed by specialized agents: a drug discovery pipeline (Figure 1A) and a scientific manuscript generation pipeline (Figure 1B).

**Agent-Driven Target Identification and Validation**    The discovery process begins with mining scientific abstract to discover novel therapeutic targets. An **Abstract Mining Agent** was prompted to identify novel drug targets for Alzheimer's Disease. It formulated and executed a complex Boolean query against the PubMed database, combining MeSH terms for the disease with keywords for drug targets (e.g., "therapeutic target," "pathway"), protein types, and modes of action (e.g., "inhibitor," "modulator") to surface relevant scientific abstracts via the NCBI E-utilities API [1]. This agent performs an initial extraction of potential protein targets. Subsequently, a **Target Evaluator Agent**, equipped with a web search tool, assesses each candidate. This agent is prompted to score each protein from 0 to 1 based on three criteria: **novelty** (is the target emerging or well-established?), **evidence** (strength of the scientific literature related to the target and the disease), and **confidence** (the agents confidence on the protein to be a novel and effective target). This structured evaluation yields a ranked list of targets for the next stage.

**Automated Predictive Model Construction**    For each validated target, the **Planner Agent** plans for ML training and delegates the task to the Coding Agent. Besides, it constructs a suite of predictive models for key ADME/Tox (Absorption, Distribution, Metabolism, Excretion, and Toxicity) properties. The agent assembles training datasets from the ChEMBL database [2] for properties and the key ADME/Tox properties, e.g., hERG inhibition, aqueous solubility, and microsomal half-life was downloaded from Therapeutic Data Commons [6]. The workflow involves feature extraction of the molecules into Morgan Fingerprint (radius=2, nBits=2048) using RDKit [3] and training an XGBoost model [4] for each endpoint. The models are evaluated using a dataset split of 72% training, 8% validation, and 20% testing across five random seeds, with comprehensive performance metrics reported in the Results section.

**Iterative *De Novo* Molecule Design and Filtering**    A **Molecule Generation Engine** utilizing the NVIDIA MolMIM API [5], a generative LLM, to produce novel chemical structures. In each cycle, the agent prompts the model to generate a batch of 30 molecules using a known active inhibitor from ChEMBL as a seed. This seed was chosen by the **Medicinal Chemist Agent** from a pool of inhibitors proven in wet-lab testing, reported in the ChEMBL library. The generated molecules are then passed to a **Molecule Evaluation Agent**, which applies the pre-trained ADME/Tox models in a **threshold-based filtering cascade**. Rather than using a composite score, a molecule must sequentially pass predefined thresholds for hERG toxicity, aqueous solubility, metabolic stability, and finally, on-target potency to be considered a hit. The highest-scoring hit from a cycle can be used as the seed for the subsequent generation cycle, iteratively refining the candidates towards a desired multi-property profile.

**Agentic Framework for Manuscript Generation**    In parallel with the scientific discovery, we utilized a conceptual agentic framework to produce this manuscript, as depicted in Figure 1B. A central **Planner Agent** (*gemini-2.5-pro*) structured the paper and delegated tasks. A **Writer Agent** (*gemini-2.5-pro*) drafted the text based on our experimental notebooks, results from the experiements and diagrams. A hierarchical **Coder Agent** managed computational tasks, delegating simple jobs to an IDE-based agent and complex coding or visualization tasks to a more powerful model (*Claude-opus-4.1*). An **Evaluator Agent** (*gemini-2.5-pro*) acted as a supervisor to refine the content. It is important to note that this writing framework was not implemented as a single, autonomous program but was actualized through a sequence of manual, iterative interactions with the specified AI models, simulating the described agentic workflow.

# 4    Results and Discussion

To evaluate our framework, we focused on Alzheimer's Disease (AD) as a case study. The Abstract Mining Agent identified five promising protein targets from the literature: SGLT2, CGAS, SEH, HDAC, and DYRK1A, each implicated in various pathways relevant to AD pathology. We then constructed classification models for these targets and deployed them in our design loop to generate novel inhibitors. This section details the datasets, model performance, and the properties of the generated molecules.

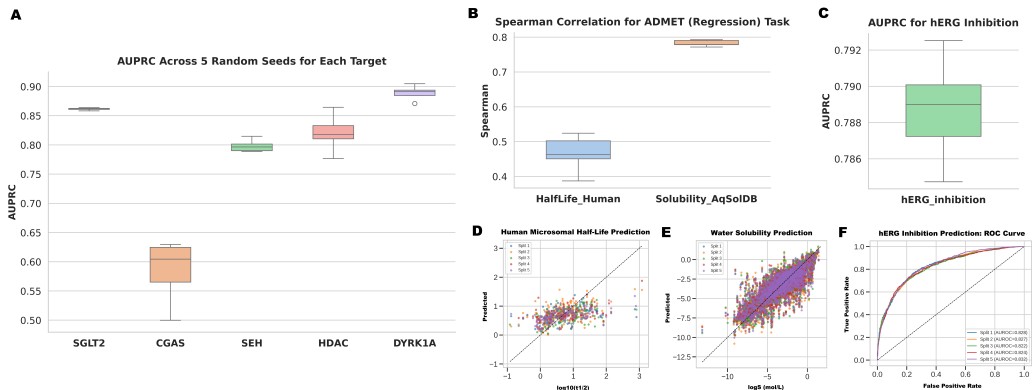

Figure 2: **Cross-task performance and robustness across five independent runs. A.** Target-specific bioactivity classification: distribution of AUPRC over five random seeds for SGLT2, CGAS, SEH, HDAC, and DYRK1A assays. Performance is highest and most stable for DYRK1A and SGLT2, moderate for SEH and HDAC, and lower with larger variance for CGAS. **B.** Spearman correlations for ADMET-related regression tasks are human microsomal half-life and aqueous solubility. Water solubility demonstrates strong alignment between predictions and ground truth where half-life is not; confirmed in **(D-E)**. (C) AUPRC distribution for hERG inhibition prediction, highlighting stable model performance across runs. (F) ROC curves for hERG inhibition classification across five independent splits, indicating consistently high AUROC (0.82) and generalization capability. Collectively, these results illustrate the agent's capability of planning and coding effective ML solutions on both classification and regression tasks critical to drug discovery.

## 4.1   Dataset Curation and Predictive Model Performance

Effective *in silico* screening requires high-quality predictive models. We curated datasets for five selected protein targets from ChEMBL, converting continuous $pIC_{50}$ values into a binary classification task of active vs. inactive. A standard $pIC_{50}$ threshold of 6.0 often proposed by automated agents resulted in severe class imbalance, leading to poorly calibrated models despite the use of class weighting during training. To address this, we manually curated the activity threshold for each target to create more balanced datasets suitable for training robust classifiers. Table 1 summarizes the final dataset composition for each target after applying the optimized thresholds. These balanced datasets formed the basis for training the XGBoost classifiers used in the subsequent molecule evaluation phase. The detailed performance metrics (e.g., AUC-ROC, F1-Score) for each classifier are presented in the repository.

Table 1: Dataset composition for bioactivity classification models. The $pIC_{50}$ threshold for each target was manually adjusted to mitigate severe class imbalance and improve model training.

| Target | AI Agent Thresholding (Pos / Neg) | Manual Thresholding (Pos / Neg, Threshold) |
|---|---|---|
| SGLT2 | 1249 / 188 | 638 / 799 (7.8) |
| CGAS | 100 / 146 | 100 / 146 (6.0) |
| SEH | 1989 / 427 | 1032 / 1384 (7.8) |
| HDAC | 926 / 730 | 696 / 960 (6.5) |
| DYRK1A | 2061 / 703 | 1211 / 1553 (7.2) |

The performance of the resulting classifiers for both target inhibition and key ADME/Tox properties is summarized in Figure 2. Our models demonstrated strong predictive accuracy for four of the five protein targets—SGLT2, SEH, HDAC, and DYRK1A—achieving median AUPRC scores ranging from 0.80 to 0.89 with 5 random (Fig. 2A). This indicates a high-capacity to correctly distinguish active from inactive compounds for these targets. However, the model for CGAS inhibition yielded an AUPRC of only 0.60, a performance level slightly better than random chance (Fig. 2A). This failure is likely attributable to the limited and heterogeneous nature of publicly available data for CGAS with

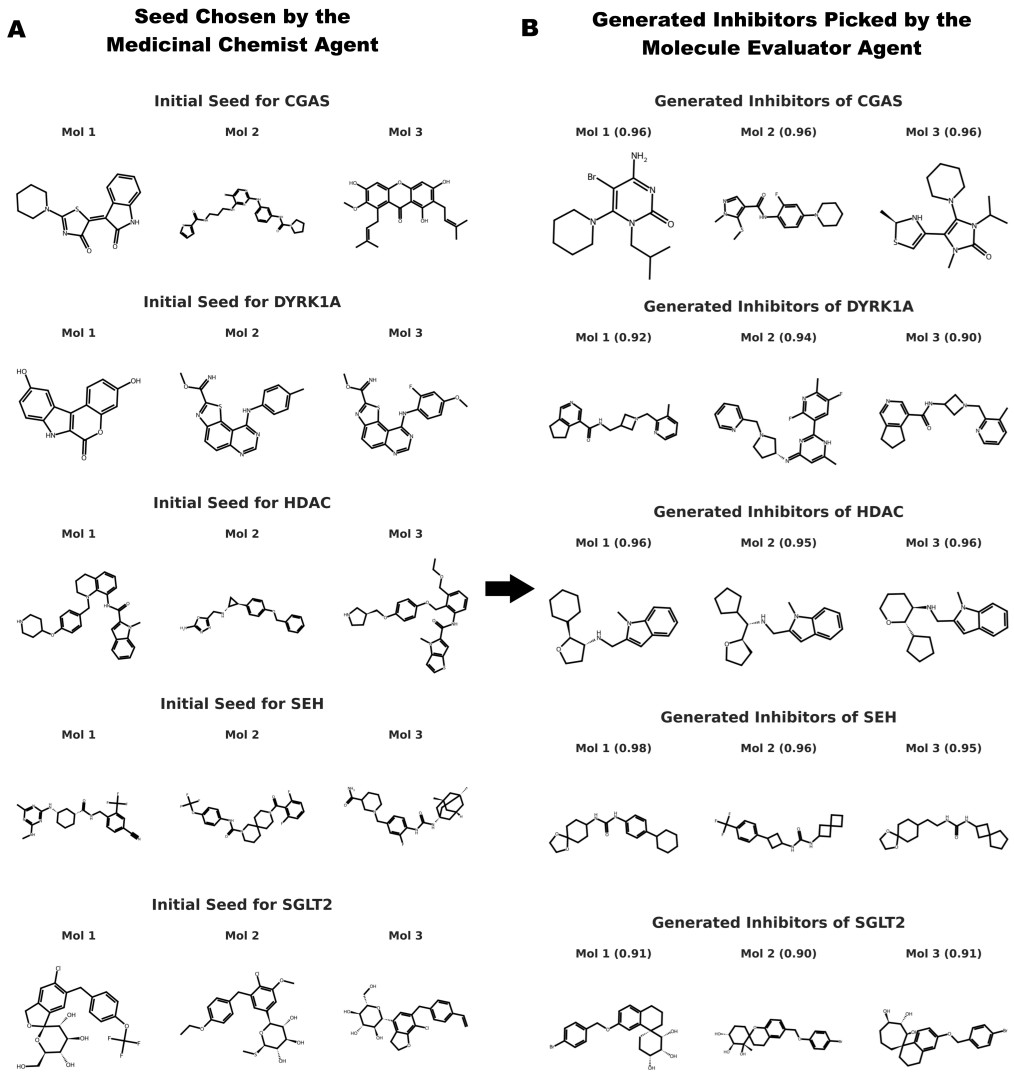

Figure 3: **Iterative *De Novo* Design of Potent Inhibitors via Agent-Driven Generation and Evaluation.** (A) A pool of known inhibitors for each target (CGAS, DYRK1A, HDAC, SEH, SGLT2), from which the Medicinal Chemist Agent selects three seeds to initiate the generative drug design process. (B) Top three novel molecules produced by the Molecule Generation Engine and successfully passed through the ADME/Tox and potency filtering cascade managed by the **Molecule Evaluator Agent**. The parenthetical value represents the predicted probability of bioactivity from the target-specific XGBoost classifier. The generated molecules often possess chemical scaffolds distinct from their seeds while retaining high predicted potency, demonstrating the framework's capacity for chemical space exploration.

Table 2: Machine Learning Predictions for Physicochemical and Safety Properties of Selected Compounds

| Target | SMILES | Predicted Solubility ($log_{10}$(mol/L)) | Predicted Half-Life ($log_{10}$(hours)) | Predicted hERG Inhibition |
|---|---|---|---|---|
| SGLT2 | O[C@@H]1[C@H](O)CC[C@]2(CCCc3ccc(OCc4ccc(Br)cc4)cc32)[C@@H]1O | -3.99 | 0.65 | Yes |
| | C[C@@]1(O)[C@@H](O)[C@H](O)CCC12CCc1cc(COc3ccc(Br)cc3)ccc1O2 | -4.11 | 0.65 | No |
| | O[C@H]1[C@H](O)CCC[C@@]2(CCCc3cc(OCc4ccc(Br)cc4)ccc32)[C@@H]1O | -4.45 | 0.65 | Yes |
| CGAS | CC(C)Cn1c(N2CCCCC2)c(Br)c(N)nc1=O | -3.68 | 0.64 | No |
| | CSc1c(C(=O)Nc2ccc(N3CCCCC3)cc2F)cnn1C | -4.20 | 0.55 | No |
| | CC(C)n1c(N2CCCCC2)c(C2=CS[C@@H](C)N2)n(C)c1=O | -3.28 | 0.44 | No |
| SEH | O=C(Nc1ccc(C2CCCCC2)cc1)NC1CCC2(CC1)OCCO2 | -4.05 | 0.84 | Yes |
| | O=C(NC1CC(c2ccc(C(F)(F)F)cc2)C1)NC1CC2(CCC2)C1 | -4.06 | 0.80 | No |
| | O=C(NCCC1CCC2(CC1)OCCO2)NC1CC2(CCCC2)C1 | -2.56 | 0.78 | No |
| HDAC | Cn1c(CN[C@@H]2CCO[C@H]2C2CCCCC2)cc2ccccc21 | -3.22 | 1.09 | No |
| | Cn1c(CN[C@@H](C2CCCC2)[C@H]2CCCO2)cc2ccccc21 | -2.88 | 1.09 | No |
| | Cn1c(CN[C@@H]2CCCO[C@H]2C2CCCC2)cc2ccccc21 | -3.10 | 1.09 | No |
| DYRK1A | Cc1cccnc1CN1CC(CNC(=O)c2cncc3c2CCC3)C1 | -2.21 | 0.68 | No |
| | Cc1cc(=N[C@@H]2CCN(Cc3ccccn3)C2)nc(-c2cc(F)c(C)nc2F)[nH]1 | -4.09 | 0.76 | No |
| | Cc1cccnc1CN1CC(NC(=O)c2cncc3c2CCC3)C1 | -1.98 | 0.68 | No |

only 100 positive and 146 negative data points where the other targets have thousands (see Table 1), rendering the 2D descriptor-based approach insufficient to capture its complex structure-activity relationship.

For the ADME/Tox properties, the ML models for Water Solubility (LogS) and hERG inhibition were highly accurate, with Spearman and AUPRC scores of 0.78 and 0.79, respectively (Fig. 2B-C, E-F). These robust models are critical for the sequential filtering cascade, enabling the effective removal of molecules with poor solubility or potential for cardiotoxicity. In contrast, the model for human microsomal half-life ($T_{1/2}$) failed, achieving an AUC-ROC of only 0.46 (Fig. 2B and D). Predicting metabolic stability is a notoriously challenging endpoint, often dependent on complex 3D conformational effects and specific metabolic pathways that are not well-represented by 2D topological descriptors alone. Given these results, the microsomal half-life models were deemed unreliable and were excluded from the subsequent generative design and filtration phase.

## 4.2 From Seed to Hit: Generative Design and Multi-Property Filtering

With a suite of validated predictive models for bioactivity and ADMET/Tox properties, we deployed our agentic framework to perform *de novo* molecule design. This phase operationalizes the core drug discovery loop, integrating molecule generation with multi-parameter evaluation to identify novel and promising hit candidates for each therapeutic target. The process is a collaborative effort between the **Medicinal Chemistry Agent**, the **Molecule Generation Engine**, and the **Molecule Evaluation Agent**.

The design cycle begins with the **Medicinal Chemistry Agent**, which selects appropriate starting points, or seed molecules, for the generative process. As shown in Figure 3A, this agent was tasked with querying the ChEMBL database to identify a small pool of known, potent inhibitors for each of the five targets. These seeds are not merely random structures but are validated chemical starting points that anchor the generative model in a region of relevant chemical space. For each target, three such seeds were selected to initiate parallel design cycles, providing diverse starting points for exploration.

For each seed, the **Molecule Generation Engine**, powered by NVIDIA MolMIM API [5], was prompted to generate a batch of 30 novel molecular analogues. This step creates a focused library of compounds that are structurally related to the known active seed. These newly generated molecules were then passed to the **Molecule Evaluation Agent**, which executes a critical, sequential filtering cascade. This cascade applies the predictive models detailed in Section 3.1 to triage candidates based on a hierarchy of essential drug-like properties. We presented the predicted properties for all the selected inhibitors in Table 4.1.

The model for human microsomal half-life was intentionally excluded from the filtering cascade due to its insufficient predictive power (Figure 2B, D), which would have added noise rather than value. Though its predictions are shown in Table 4.1, this decision highlights the framework's logic of only using models that meet a required performance threshold.

The results, summarized in Figure 3B, show the framework consistently produced novel molecules with high predicted bioactivity scores, often exceeding 0.90. A key success was its ability to perform scaffold hopping—generating molecules with core structures distinct from the initial seed, as seen when a complex HDAC seed inspired simpler, potent indole-based scaffolds. However, for the CGAS target, the low reliability of the bioactivity model (AUPRC=0.60) renders the generated hits highly speculative. This outcome confirms a core principle: the quality of generated candidates is fundamentally limited by the predictive models guiding the design. The process demonstrates a powerful, automated cycle for rapidly identifying diverse and high-quality chemical matter.

## 5  Conclusion

This work demonstrates the application of a multi-agent framework to accelerate the early-stage drug discovery pipeline, from target identification to the generation of novel hit compounds. Our integrated system successfully produced unique molecular scaffolds with high predicted potency and favorable ADME/Tox profiles for multiple protein targets by synergizing a generative molecular model with a cascade of custom-trained predictive classifiers. However, our findings also underscore the critical limitations of current autonomous systems. The failure to build reliable models for data-scarce targets like CGAS or for complex endpoints like microsomal half-life, coupled with the necessity of manual expert intervention to curate balanced datasets, reveals that the success of such frameworks is fundamentally tethered to the quality and availability of underlying data.

These results suggest that while AI agents are powerful tools for navigating vast chemical space and automating high-throughput evaluation, they currently operate most effectively within a "human-in-the-loop" paradigm where scientific expertise guides data curation, model validation, and strategic decision-making. Future work should focus on enhancing model robustness by incorporating more sophisticated features beyond 2D descriptors and, most critically, on closing the discovery loop through prospective experimental validation. The synthesis and biological testing of the most promising candidates generated in this study will be the ultimate measure of the framework's ability to translate *in silico* potential into tangible therapeutic progress.

## 6  AI Agent Setup

The research framework was implemented using a multi-agent system where different Large Language Models (LLMs) and specialized tools were assigned to distinct roles. The orchestration was tailored to two main workflows: drug discovery and manuscript generation.

**Large Language Models and Agent Roles.**  The system primarily utilized Google's Gemini family of models. **Gemini 2.5 Pro** was employed for high-level tasks requiring complex reasoning, including the roles of the central **Planner Agent**, the **Writer Agent**, and the **Evaluator Agent**. For the hierarchical Coder Agent, which required advanced code generation, **Claude-opus-4.1** was used. The manuscript's workflow diagram (Figure 1) was generated using the **nano-banana** image generation model, from a hand-drawn sketch.

**Orchestration and Workflow.**  The multi-agent collaboration was not a single autonomous program but was orchestrated through a sequence of iterative, manual interactions that simulated the conceptual framework.

- **Drug Discovery Workflow:** This pipeline followed a modular, sequential orchestration. It began with an Abstract Mining Agent that autonomously queried PubMed. Its findings were then passed to a Target Evaluator Agent for scoring. For validated targets, a Medicinal Chemist Agent selected seed molecules from the ChEMBL database. These seeds were fed into the Molecule Generation Engine, which then handed off the novel structures to a Molecule Evaluation Agent for filtering.

- **Manuscript Generation Workflow:** This process was managed by a central Planner Agent (Gemini 2.5 Pro) that structured the paper and delegated tasks to specialized agents for writing, coding, and evaluation, simulating a collaborative, human-in-the-loop scientific writing process.

**Tool Integration.** The agents were equipped with a suite of external tools to perform specialized scientific tasks:

- **Literature Mining:** The Abstract Mining Agent used the **NCBI E-utilities API** to execute complex boolean queries against the **PubMed** database.
- **Cheminformatics and Data:** Agents accessed the **ChEMBL** database for inhibitor data and the **Therapeutic Data Commons** for ADME/Tox datasets. Molecular feature extraction was performed using the open-source RDKit library.
- **Predictive Modeling:** The framework trained and utilized custom-trained XGBoost models for bioactivity and ADME/Tox predictions.
- **Generative Chemistry:** The de novo design of molecules was performed using the **NVIDIA MolMIM API**, which functioned as the core Molecule Generation Engine.

This setup created a synergistic system where the reasoning and planning capabilities of LLMs were combined with the domain-specific power of external scientific databases and computational tools.

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
