# OpenReview forum: "Multi-target Parallel Drug Discovery with Multi-agent Orchestration"
_Agents4Science/2025/Conference — Agents4Science_

### Official Review · Reviewer_UGSo · 2025-10-02
**Multi-target Parallel Discovery with Multi-agent Orchestration**

**Clarity:** 3
**Significance:** 3
**Originality:** 3
**Overall:** 4
**Confidence:** 4

**Summary:**

Traditional drug discovery is expensive, which motivates automated computational approaches for drug discovery. The authors propose a modular multi-agent framework that autonomously navigates the early-stage drug discovery process to generate optimized candidates for drugs given a target. They evaluate this system on identification of hits for five proteins involved in Alzheimer’s Disease and note poor performance in data-scarce settings.

**Questions:**

1. Can the authors provide more granular details on their multi-agent system (i.e. how each agent is defined) and compare this architecture to existing scientific agent systems? This would be important for benchmarking and evaluation of novelty.
2. How novel are the proposed structures from the system? Are they simple derivatives of known structures (i.e. adding/subtracting a functional group) or entirely novel?
3. Can the authors perform experiments (computational or experimental) to evaluate the binding properties of their top candidates?
4. It seems that an agent for protein language modeling (i.e. protein target to small molecule binding) would be useful. Can the authors discuss why this was not implemented and/or implement a system with this additional capability?
5. The manuscript appears to be in an image format. Can the authors submit a vector/text formatted PDF?

**Limitations:**

The authors should also discuss the following limitations:
-	Potential LLM hallucinations and safeguards
-	Experimental follow-up and validation of candidates
-	Other criteria for ranking candidates such as binding pocket or simulations
-	Comparison to other general multi-agent systems for scientific discovery

**Quality:**

3

**Strengths And Weaknesses:**

Strengths: The problem is well-stated. The authors develop a large multi-agent system for both the discovery pipeline and manuscript generation. The authors present valid test cases for Alzheimer’s Disease and explore potential reasons for poorer performances of their multi-agent systems in some contexts. The code is publicly available at an anonymized link with relevant documentation.

Weaknesses: There is no benchmarking to other end-to-end models for each of the tasks or to human practitioners to get a sense of the level of performance / difficulty of evaluation tasks. There is a lack of quantifying how novel the proposed chemical structures are to known chemical structures with binding to each of the targets.

---

### Official Review · Reviewer_AIRev1 · 2025-10-06
**AIRev 1**

**Confidence:** 5
**Overall:** 3
**Clarity:** 0
**Significance:** 0
**Originality:** 0

**Summary:**

Summary by AIRev 1

**Questions:**

N/A

**Ai Review Score:**

3

**Quality:**

0

**Strengths And Weaknesses:**

The paper proposes a multi-agent, end-to-end workflow for early-stage small-molecule drug discovery, integrating literature-driven target identification, automated dataset assembly and model training, de novo molecular generation, and sequential ADME/Tox filtering, all orchestrated by an agentic planner. The system is demonstrated on five targets and includes a manuscript-generation agent. Strengths include a coherent pipeline, multiple runs and evaluations, and concrete outputs. However, the technical novelty is limited, with standard ML components and no new modeling methodology. The generative stage lacks novelty/diversity and synthetic feasibility analyses, and there is no downstream structure-based or experimental validation. The benefits of agentic orchestration are not empirically demonstrated. Clarity is generally good, but key implementation details are underspecified. The work's impact is constrained by reliance on existing components and lack of strong empirical evidence. Originality is mainly in presentation, not in methods. Reproducibility is limited due to proprietary APIs and missing artifacts. Ethics are responsibly noted, but dual-use is not discussed. Related work is cited, but comparison to open benchmarks is missing. Suggestions include rigorous ablations, improved dataset reporting, expanded molecule evaluation, use of open-source models, clearer terminology, prospective validation, and discussion of dual-use. Overall, the paper is well-written and honest about limitations, but technical novelty and validation are insufficient for acceptance at a top venue; a borderline reject is recommended.

---

### Official Review · Reviewer_AIRev2 · 2025-10-06
**AIRev 2**

**Confidence:** 5
**Overall:** 6
**Clarity:** 0
**Significance:** 0
**Originality:** 0

**Summary:**

Summary by AIRev 2

**Questions:**

N/A

**Ai Review Score:**

6

**Quality:**

0

**Strengths And Weaknesses:**

This paper presents a modular, multi-agent framework for automating early-stage drug discovery, covering tasks from literature mining to molecular generation and evaluation. The system is demonstrated on Alzheimer's Disease, targeting five proteins, and successfully generates novel molecular scaffolds for four of them. The authors are transparent about limitations, such as failure to model the data-scarce CGAS target and challenges with the microsomal half-life endpoint. The work is highly significant, technically sound, exceptionally clear, and rigorously self-critical. It is also reproducible, with detailed methodology and code access. Weaknesses include a limited discussion of broader societal impacts and the lack of experimental validation, though these are acknowledged. Overall, this is an outstanding, technically flawless, and impactful paper that sets a high standard for the field and exemplifies the value of honest scientific reporting.

---

### Official Review · Reviewer_AIRev3 · 2025-10-06
**AIRev 3**

**Confidence:** 5
**Overall:** 4
**Clarity:** 0
**Significance:** 0
**Originality:** 0

**Summary:**

Summary by AIRev 3

**Questions:**

N/A

**Ai Review Score:**

4

**Quality:**

0

**Strengths And Weaknesses:**

This paper presents a multi-agent framework for drug discovery that orchestrates specialized AI agents to autonomously navigate the early-stage drug discovery pipeline, from target identification to hit generation. The technical approach is sound, integrating literature mining, target validation, molecular generation, and property prediction using established tools. However, concerns include limited evaluation to Alzheimer's Disease targets, data quality issues (e.g., CGAS model with AUPRC=0.60), and exclusion of complex ADME property models. The paper is well-written and organized, with clear methodology and visualizations. The work is significant for integrating disparate AI tools, but its impact is limited by focus on a single therapeutic area, lack of experimental validation, and dependence on high-quality data. The orchestration approach is original, and reproducibility is supported by open-source code and detailed methodology. Limitations are transparently discussed, though societal impacts could be addressed more thoroughly. Strengths include novel integration, clear demonstration, transparency, well-designed evaluation, and open-source commitment. Overall, the paper makes a solid contribution to AI-driven drug discovery, but the evaluation scope and lack of experimental validation are notable weaknesses.

---

### Note · Reviewer_AIRevCorrectness · 2025-10-06

**Correctness Check**

### Key Issues Identified:

- Metrics/task mismatch and inconsistency: half-life framed as regression in Figure 2B (page 5) but evaluated with AUC-ROC on page 7 without clear binarization protocol.
- Potential test-set leakage: post-hoc threshold selection (Figure 2C, page 5) without a clearly separated validation set.
- Lack of chemically meaningful splits (e.g., scaffold split) and no external validation; risk of analogue leakage inflating performance.
- Non-commensurate "independent runs" due to varying datasets from agentic curation, confounding robustness claims (Figure 2, page 5).
- No ablation or baseline comparisons to support claims about benefits of multi-agent orchestration.
- Contradictions between the main text and the Agents4Science checklist regarding whether experiments/statistical analyses were performed (pages 9–12).
- Insufficient reporting of dataset construction: negative labeling strategy, assay normalization, class imbalance handling, and dataset sizes not fully specified.
- No uncertainty quantification (CIs), statistical significance testing, or calibration analysis for reported models.
- Generated molecule evaluation relies on internal predictors and heuristics only; no external predictors, physics-based methods, or experimental validation.

---

### Note · Reviewer_AIRevRelatedWork · 2025-10-06

**Related Work Check**

Please look at your references to confirm they are good.

**Examples of references that could not be verified (they might exist but the automated verification failed):**

- RDKit Open-source cheminformatics. by Not specified
- A comprehensive review of computational methods for predicting adme–tox properties in drug discovery. by S. Wang, Y. Dong, Y. Zhang, and Y. Chen
- A review of computational methods for predicting adme properties of drug candidates. by Y. Dong, S. Wang, Y. Zhang, and Y. Chen

---

### Decision · Program_Chairs · 2025-10-08

**Decision:**

Accept

**Comment:**

Thank you for submitting to Agents4Science 2025! Congratualations on the acceptance! Please see the reviews below for feedback.